# Decline of adolescent smoking in Ireland 1995–2015: trend analysis and associated factors

Shasha Li, Sheila Keogan, Keishia Taylor, Luke Clancy

## ABSTRACT

**Objectives** The study examines trends in smoking among Irish adolescents aged 15–16 years between 1995 and 2015 and the factors associated with their smoking behaviours between 2007 and 2015.

**Methods** Data were obtained from the European School Survey Project on Alcohol and Other Drugs Ireland between 1995 and 2015. To examine the gender gap, two-sample proportion tests were used. Multivariate logistic regression was performed to examine the factors associated with smoking behaviours. Dependent variable is whether a respondent is a smoker in last 30 days. Independent variables include gender, survey years, perceived ease of access to cigarettes, perceived risk of smoking, perceived relative wealth, parental monitoring, maternal relationship, family structure, truancy and peer smoking.

**Results** Smoking prevalence has dropped from 41% in 1995 to 13% in 2015. The prevalence was much higher among girls than boys in 1995. The gender gap was closed by 2015. Multivariate regression results show that peer smoking, perceived access to cigarettes, perceived risks of smoking, parental monitoring, truancy, maternal relationship, perceived relative wealth and family structure were all significantly associated with adolescent smoking, and some of the factors had different effects for female and male students.

**Conclusion** Ireland has successfully achieved a considerable decrease of adolescent smoking from 1995 to 2015, during which various tobacco control policies have been implemented. In addition, the gender gap on adolescent smoking has been closed during the period. Adolescent smoking could be further improved through strengthening enforcement on adolescent access to cigarettes and maintaining a high-intensity tobacco control media campaign targeting adolescents. Parents could also contribute by enhancing monitoring.

## Strengths and limitations of this study

► The data are from the best available surveys on Irish adolescents from 1995 to 2015 using an internationally validated survey instrument, including comprehensive measurement for smoking-related factors. The datasets in different years are comparable in data collection and questions.
► The study is based on multivariable logistic regression which controls for confounders.
► The factors associated with the changing gender gap in adolescent smoking were examined.
► Other factors potentially related to adolescent smoking were not included in the surveys, such as parental smoking.
► Limited data availability for some of the European School Survey Project on Alcohol and Other Drugs waves are acknowledged.

be prioritised in all of the initiatives outlined in the policy.[3]

The parties to the 2003 WHO Framework Convention on Tobacco Control stated their 'deep concern' regarding tobacco consumption by children and adolescents, emphasising price and tax measures as effective means of reducing tobacco consumption among young people.[4] Between 1995 and 2015, the real retail price per package of 20 cigarettes in Ireland has been increased almost every year (see online supplementary appendix I). The European Tobacco Products Directive (2014/40/EU) places restrictions on the use of tobacco packaging designed to appeal to children and prohibits flavoured cigarettes and roll-your-own tobacco.[5] In Ireland, stronger legislation regarding the complete standardisation of tobacco packaging came into force in September 2017 and should reduce further the effect of tobacco advertising on adolescents.[6]

Between 1995 and 2015, a series of national-level tobacco control policies were introduced (online supplementary appendix II), although there were no school-specific tobacco control policies implemented in

## INTRODUCTION

People who take up smoking at a younger age become more dependent and find it harder to quit than smokers who start later in their lives,[1] [2] so policies designed to discourage adolescents from starting to smoke have been at the forefront of tobacco prevention in recent years. In Ireland, the Tobacco Free Ireland (TFI) report of 2013 stated that the protection of children must

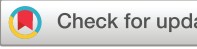

TobaccoFree Research Institute Ireland, Dublin, Ireland

**Correspondence to**
Dr Shasha Li; shashali@tri.ie

Ireland. For example, the implementation of age limit law under Public Health (Tobacco) Act 2002 was not fully implemented until April 2007. It is an offence to sell cigarettes or other tobacco products to persons aged under 18 years. In addition, since 2009, retailers are required to register with the National Tobacco Control Office. Vending machines were banned except in licensed establishments.

There has been a large volume of research conducted regarding interventions against adolescent smoking, including studies evaluating policies to restrict access and raise awareness of risk, with mixed results.[7–9] Some studies have examined perceptions of risk and its association with smoking and the majority reported a negative association between risk perception and smoking.[10 11] Other studies have investigated correlates in the domestic and social sphere, including associations with parental monitoring, relationships with parents, family structure, truancy from school and peer smoking. [12-16] It is also established that adolescents from a lower socioeconomic background are more likely to smoke.[17 18] This study first establishes the declining trends in smoking among Irish people aged 15 and 16 years between 1995 and 2015 and the closing of the gender gap. Second, we examine potential factors associated with their smoking behaviours using the 2007–2015 surveys. Based on the factors mentioned in existing literature and taking account of availability in the dataset, we explored the relationship to smoking prevalence of gender, survey years, perceived ease of access to cigarettes, perceived risk of smoking, perceived relative wealth, parental monitoring, maternal relationship, family structure, truancy and peer smoking, and also discussed the changes of the factors over time. We cannot assess the effects of the use of electronic (e)-cigarettes on trends using these data as we only have data on e-cigarette from the 2015 survey and they were only introduced in 2013. We have expressed our fears concerning e-cigarette and the need for close monitoring elsewhere.[19 20]

## METHODS

### Data source and sample

This study used data from the European School Survey Project on Alcohol and Other Drugs (ESPAD) Ireland. The main purpose of the survey was to collect comparable data on substance use among 15-year-old and 16-year-old students across Europe, in order to monitor trends within and between countries, including Ireland.[21] ESPAD Surveys were conducted every 4 years between 1995 and 2015, resulting in six waves of data from 26 countries, and 35 countries participating in 2015.

The sampling procedures, data collection and questionnaire used in Ireland were consistent with the international ESPAD study protocol.[19] School students born in specific calendar years were eligible and selected using stratified random sampling. Data were collected anonymously through paper-and-pencil, self-completion questionnaires administered in the classroom. After

**Table 1** Sample sizes, gender and response rates of the European School Survey Project on Alcohol and Other Drugs Ireland Surveys (1995–2015)

| Years | 1995 | 1999 | 2003 | 2007 | 2011 | 2015 |
|---|---|---|---|---|---|---|
| Sample size | 1849 | 2277 | 2407 | 2221 | 2207 | 1470 |
| Male (%) | 49 | 49 | 51 | 45 | 50 | 51 |
| Response rate (%) | 96 | 92 | 96 | 94 | 94 | 86 |

standardised cleaning procedures, the datasets were obtained from the ESPAD official database. Full accounts of the methodology of the study in each survey year could be found in the respective reports of the ESPAD project.[21-23]

Smoking prevalence for all six ESPAD waves is available and used for the trend analysis. The raw 1999 and 2003 survey datasets are unavailable and the 1995 dataset did not include most of the measures used in the study. Therefore, only the 2007, 2011 and 2015 studies were used to assess the associated factors of smoking. Sample characteristics are reported in table 1.

### Measures

Respondents were asked how frequently they had smoked in the last 30 days, with answers ranging from 'not at all' to 'more than 20 cigarettes per day'. Those who answered 'not at all' are non-smokers and those who had smoked at least once in the last 30 days are smokers. Current or 30-day smoking prevalence rate is the proportion of smokers.

The questionnaire included items about respondents' awareness of and experience with cigarette smoking, perceived family wealth, parental monitoring, relationship with parents, family structure, truancy and peer smoking.

Students were asked how difficult it would be to get cigarettes if they wanted them and to what extent people risk harming themselves (physically or in other ways) if they smoke occasionally. The majority of students thought it would be easy to get cigarettes and about half of the students perceived a moderate/great risk from smoking occasionally.

Socioeconomic status was estimated by how well off students perceived their family to be compared with other families on four points from 'much better off' to 'less well off'. Respondents were also asked whether their parents know where they spend Saturday nights (always, quite often, sometimes or usually do not know) and whether they were satisfied with their relationship with their mother. Students also listed the members of their household and around 14% of the students were from one-parent families.

Respondents were asked about truancy by reporting the number of days on which they had skipped one or more days during the last 30 days. In addition, they were asked how many of their friends smoked cigarettes.

**Table 2** Summary of the results and changes in key measures associated with smoking in Irish European School Survey Project on Alcohol and Other Drugs (ESPAD) Surveys 2007–2015

| | ESPAD year | | |
| --- | --- | --- | --- |
| | 2007 | 2011 | 2015 |
| **Access to cigarettes*** | | | |
| Difficult (%) | 12.0 | 16.2 | 28.1 |
| Easy (%) | 41.7 | 41.9 | 42.4 |
| Do not know (%) | 46.3 | 41.9 | 29.5 |
| **Risk of smoking cigarettes occasionally** | | | |
| No/slight risk (%) | 39.6 | 43.0 | 41.6 |
| Moderate/great risk (%) | 58.0 | 54.2 | 55.8 |
| Do not know (%) | 2.3 | 2.8 | 2.5 |
| **Perceived family relative wealth*** | | | |
| Much better off (%) | 15.8 | 10.8 | 15.8 |
| Better off (%) | 32.4 | 27.7 | 25.8 |
| About the same (%) | 46.2 | 56.2 | 48.6 |
| Less well off (%) | 5.5 | 5.3 | 9.8 |
| **Parents knowing where students spend Saturday nights*** | | | |
| Know always (%) | 47.4 | 49.6 | 62.7 |
| Know quite often (%) | 30.0 | 30.3 | 23.3 |
| Know sometimes (%) | 15.4 | 14.9 | 8.9 |
| Usually do not know (%) | 7.2 | 5.3 | 5.1 |
| **Relationship with mother*** | | | |
| Satisfied (%) | 83.4 | 87.4 | 87.4 |
| Neither nor (%) | 7.8 | 5.0 | 5.2 |
| Not satisfied (%) | 8.8 | 7.5 | 7.3 |
| **One-parent family** | | | |
| Two or more parents (%) | 87.0 | 86.1 | 86.0 |
| One parent (%) | 13.0 | 13.9 | 14.0 |
| **Skipping school in the last 30 days*** | | | |
| None (%) | 73.6 | 80.4 | 80.2 |
| 1–4 days (%) | 21.2 | 16.7 | 16.1 |
| 5 days+ (%) | 5.2 | 2.9 | 3.8 |
| **Peers smoking*** | | | |
| None (%) | 12.4 | 11.9 | 33.5 |
| A few/some (%) | 67.6 | 69.7 | 55.8 |
| Most/all (%) | 20.0 | 18.4 | 10.6 |

*$\chi^2$ statistically significant at 0.05 level.

The frequencies of responses for each predictor category are shown in table 2. Most responses changed significantly across the survey years, notably in perceived access to cigarettes, parental monitoring and peer smoking. Particularly, in 2007 and 2011, only about 12% of the students reported that none of their friends smoked, but by 2015 it had increased to 34%. The proportion of students who claimed most/all of their friends smoked had decreased from 20% in 2007 to 11% in 2015. Access to cigarettes had become more difficult across the three survey waves, with students who reported that it was difficult to obtain cigarettes increasing from 12% in 2007 to 28% in 2015. More students claimed that their parents always know where they spend Saturday nights, from 48% in 2007 to 63% in 2015.

## Statistical analysis

To examine the gender gap in smoking prevalence between 1995 and 2015, two-sample proportion tests were used and p values are reported. The main analysis examined the factors associated with adolescents' smoking behaviours across the last three survey waves using

**Table 3** Thirty-day smoking prevalence (%) in Irish European School Survey Project on Alcohol and Other Drugs Surveys (ESPAD) from 1995 to 2015

| Year | 1995 | 1999 | 2003 | 2007 | 2011 | 2015 |
|---|---|---|---|---|---|---|
| Male | 36.7 | 32 | 28 | 19.3 | 18.6 | 13.1 |
| Female | 44.9 | 42 | 37 | 26.8 | 23.2 | 12.8 |
| Total | 40.9 | 37 | 33 | 23.4 | 20.9 | 13.0 |
| P values* | <0.001 | <0.001 | <0.001 | <0.001 | 0.009 | 0.83 |

*The null hypothesis is that female and male students had the same smoking prevalence.

multivariate logistic regression. The dependent variable was whether or not a student had smoked in the last 30 days. Independent variables included gender, survey years and the measures listed in table 2. The analysis was then repeated for each gender individually to detect if any factors played different roles between female and male students. All of the statistical analysis was conducted in IBM SPSS Statistics V.22.

## RESULTS
### Trend of adolescent smoking and the closing of gender gap
Thirty-day smoking prevalence among boys and girls for each survey wave is shown in table 3 and figure 1. In 1995, female students had a 30-day smoking prevalence of 44.9%, much higher than the prevalence of male students of 36.7% (p<0.001). Along the survey years, the prevalence for both genders dropped significantly, with girls achieving a greater decline. By 2015, the female and male smoking prevalence is 12.8% and 13.1%, respectively. With slightly fewer female students smoking than male students, the gender gap was closed by 2015, which is confirmed by the p value of 0.83.

### Predictors of adolescent smoking
Table 4 presents the multivariate logistic regression results taking account of the potential factors associated with adolescent smoking. Peer smoking, perceived access to cigarettes, perceived risk of smoking, parental monitoring, truancy, maternal relationship, perceived relative wealth and family structure were all significantly associated with adolescent smoking, and some of the factors had different effects for female and male students.

Peer smoking had the strongest effect. A student with a few/some friends who smoked was four times more likely to smoke than a student who had no smoking friends. If most/all their friends smoked, the odds of smoking were 27 times higher for female students and 14 times higher for male students.

Students who reported a lower risk from smoking occasionally were twice more likely to smoke than those reported greater risk from smoking. For female students, those who reported 'easy to access' cigarettes were about twice as likely to smoke as those who reported it 'difficult'. However, for male students, there was no significant difference. Interestingly, for both genders, those who reported 'do not know' if it is easy or difficult to access cigarette are about three times more likely to smoke than those who reported it as difficult.

Family appears to play an important role in adolescent smoking. Students whose parents usually do not know their whereabouts on Saturday nights are about three times more likely to smoke than the ones whose parents always know. For male students, the odds are even larger, at close to five times. For students who were not satisfied with their relationship with their mother, the odds of smoking are about two times higher than for students who were satisfied. Being from a one-parent family did not have significant effect on male smoking. However,

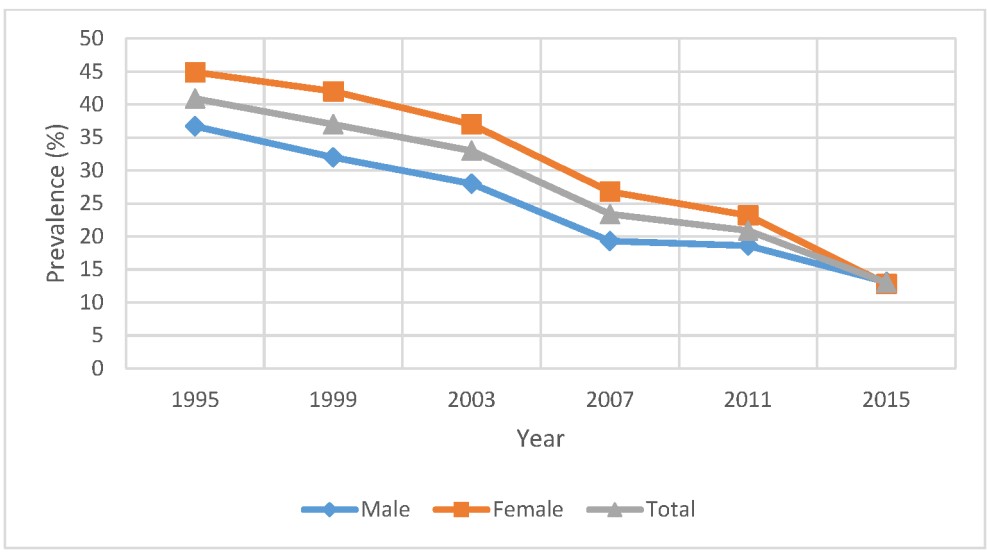

**Figure 1** Trend of adolescent smoking prevalence: 1995–2015.

**Table 4** Multivariate logistic regression results of various factors potentially associated with smoking from Irish European School Survey Project on Alcohol and Other Drugs (ESPAD) Surveys 2007–2015

| | OR (95% CI) | | |
| --- | --- | --- | --- |
| | **Total** | **Male** | **Female** |
| Gender | | | |
| Male | 1 | **NA** | **NA** |
| Female | **1.50 (1.25 to 1.80)** | **NA** | **NA** |
| ESPAD year | | | |
| 2007 | 1 | 1 | 1 |
| 2011 | 1.05 (0.85 to 1.28) | 1.20 (0.89 to 1.62) | 0.93 (0.70 to 1.23) |
| 2015 | 0.91 (0.70 to 1.18) | 1.17 (0.80 to 1.70) | 0.72 (0.50 to 1.03) |
| Access to cigarette | | | |
| Difficult | 1 | 1 | 1 |
| Easy | **1.44 (1.01 to 2.07)** | 1.12 (0.65 to 1.93) | **1.69 (1.04 to 2.74)** |
| Do not know | **2.88 (2.02 to 4.10)** | **2.54 (1.51 to 4.29)** | **3.04 (1.87 to 4.92)** |
| Perceived risk of smoking | | | |
| Great/moderate risk | 1 | 1 | 1 |
| No/slight risk | **1.90 (1.58 to 2.28)** | **1.65 (1.26 to 2.16)** | **2.16 (1.67 to 2.80)** |
| Do not know | 0.66 (0.27 to 1.59) | 0.45 (0.09 to 2.16) | 0.86 (0.28 to 2.61) |
| Perceived family relative wealth | | | |
| About the same | 1 | 1 | 1 |
| Much better off | **1.33 (1.02 to 1.73)** | 1.15 (0.78 to 1.69) | 1.42 (0.97 to 2.08) |
| Better off | 0.84 (0.68 to 1.05) | **0.68 (0.49 to 0.93)** | 1.01 (0.75 to 1.36) |
| Less well off | 1.07 (0.74 to 1.56) | 0.92 (0.52 to 1.61) | 1.20 (0.73 to 1.99) |
| Parents know where students are on Saturday nights | | | |
| Know always | 1 | 1 | 1 |
| Know quite often | **1.46 (1.18 to 1.82)** | **1.78 (1.28 to 2.47)** | 1.19 (0.88 to 1.61) |
| Know sometimes | **2.43 (1.89 to 3.13)** | **2.28 (1.58 to 3.29)** | **2.59 (1.81 to 3.70)** |
| Usually do not know | **3.24 (2.29 to 4.59)** | **4.79 (2.95 to 7.75)** | **2.04 (1.23 to 3.36)** |
| Relationship with mother | | | |
| Satisfied | 1 | 1 | 1 |
| Neither nor | 1.33 (0.96 to 1.84) | 1.36 (0.80 to 2.31) | 1.34 (0.87 to 2.06) |
| Not satisfied | **1.82 (1.36 to 2.44)** | **1.93 (1.19 to 3.14)** | **1.81 (1.24 to 2.63)** |
| Family structure | | | |
| Two parents or more | 1 | 1 | 1 |
| One parent | 1.25 (0.97 to 1.61) | 0.92 (0.62 to 1.38) | **1.57 (1.12 to 2.20)** |
| Skipping school | | | |
| None | 1 | 1 | 1 |
| 1–4 days | **1.93 (1.57 to 2.38)** | **1.99 (1.47 to 2.69)** | **1.89 (1.42 to 2.52)** |
| 5 days+ | **2.80 (1.91 to 4.11)** | **2.46 (1.40 to 4.32)** | **3.46 (2.00 to 5.99)** |
| Friends that smoke | | | |
| None | 1 | 1 | 1 |
| A few/some | **3.77 (2.32 to 6.13)** | **3.63 (1.89 to 6.98)** | **4.23 (2.02 to 8.86)** |
| Most/all | **18.85 (11.39 to 31.22)** | **14.06 (7.09 to 27.87)** | **26.81 (12.51 to 57.48)** |

Bold numbers indicate statistical significance at the 0.05 level.
NA, not applicable.

female students from a one-parent family were about twice as likely to smoke.

Truancy was also associated with 30-day smoking, with students who skipped more days off school being more likely to smoke.

Female students were more likely to smoke than male students when controlling for the listed predictors.

Perceived family relative wealth did not matter for female students. For male students who perceived their families to be 'better off' were less likely to smoke than those who answered 'about the same'. Adolescents from less well off families were not significantly more likely to smoke than those who answered about the same, when controlling for the named factors. Moreover, when including both genders, students who perceived their families to be much better off were, however, more likely to smoke than those from average families.

## DISCUSSION

This study confirms the decline in smoking among people aged 15–16 years and the closing of the gender gap. This study also supports research showing that perceived ease of access to cigarettes, lower perceived risk of smoking, lower parental monitoring, unsatisfied family relationships, truancy and peer smoking are associated with adolescent smoking.

The prevalence of smoking in the past 30 days declined between 1995 and 2015, with smoking rates among girls reducing more steeply than boys, thus closing the gender gap. However, when controlling for the variables shown in table 4, no significant change across the three survey years was found and girls had higher odds of smoking than boys did. This suggests that the factors included in the model may explain the decline in smoking prevalence. In support of this, we find that most of the factors have changed significantly in a favourable direction between data waves, including perceived access to cigarettes, parental monitoring and peer smoking.

Perceived ease of access to cigarettes decreased between 2007 and 2015. Students claiming that it was difficult to get cigarettes increased from 12% in 2007 to 28% in 2015. Several policies introduced during this period might contribute to the increase in difficulty accessing to cigarettes. In particular, the implementation of age limit law under Public Health (Tobacco) Act 2002 has officially been in force since April 2007. It is an offence to sell cigarettes or other tobacco products to persons aged under 18 years. In addition, since 2009 retailers are required to register with the National Tobacco Control Office. Vending machines were banned except in licensed establishments. These measures followed a 2007 ban on packets containing less than 20 cigarettes and a number of substantial increases to the excise duty on tobacco products, a known effective strategy to reduce prevalence among adults and children.[24]

Parents who 'know always' students' whereabouts on Saturday nights increased from 48% in 2007 to 63% in 2015. This may point to family dynamics or family culture as an important aspect of reducing adolescent smoking, although it should not be assumed that simply asking adolescents where they are going would affect their behaviour. Correlations between parental monitoring and how much caring and support they think they got may in fact be a proxy for other aspects of home and social life needing further exploration.

Peer smoking, the strongest factor for predicting adolescent smoking, has also improved between 2007 and 2015. Students with no smoking friends significantly increased from 12% to 34%, and students claiming most/all friends smoked dropped from 20% to 11%. According to the results, the odds of smoking are 27 times higher for female students who report that most/all their friends smoke than for those with no smoking friends. However, the direction of causality is not necessarily clear due to the limitation of cross-sectional study. Adolescents who smoke may seek out other smokers as friends, but equally, adolescents may imitate the behaviours of their friends, including starting to smoke. The fact that there are fewer smokers in the population may, at least partially, account for the fact that the students reported fewer friends as smokers but this would perhaps suggest a more passive or coincidental occurrence than the perceived role of peers in adolescent smoking.[25]

Despite the introduction of a number of policy measures, there was little change in perceived risk associated with smoking tobacco between 2007 and 2015. Textual health warnings on cigarette packaging were introduced in 2003 and were further expanded in 2008. Further, mandatory graphic health warnings on tobacco products were introduced in 2013, so that a text-only warning occupies at least 32% of the front and a pictorial warning occupies at least 45% of the back of the pack. These measures were intended to increase awareness of the health risks associated with smoking. However, no school-based programmes were developed in Ireland aiming to raise the awareness of risks associated with smoking during this period. The fact that perceived risk of smoking did not increase among these respondents between 2007 and 2015 suggesting that this effect was not the mode of action of existing population-wide interventions.

There is a wealth of evidence to link adolescent smoking and low socioeconomic status, but this study has found that perceived lower relative wealth was not linked to an increased likelihood of smoking. There are some explanations for the absence of effect. First, although we meant to capture socioeconomic status by perceived family wealth compared with other student, the measure is not objective, which might raise bias on the estimation of socioeconomic effect. For example, if most of a student's friends are from very wealthy families, despite the student's real family wealth, the student might feel he is less well off than them, which will make the measure far from the true socioeconomic status. Second, the association between low socioeconomic status and smoking may have been accounted for using other factors that were

in the model. However, the reasons for the finding that 'perceived much better off family wealth' was associated with higher rates of smoking may be related to increased disposable income which is known to lessen the effects of price in adults.[26][27]

The closing of the gap between females and males is not fully explained by the changes in the variables examined alone and may be due to a differential effect of the changes in legislation which will be examined in other data sources.

## CONCLUSION

Ireland has successfully achieved a considerable decrease in adolescent smoking from 1995 to 2015. In addition, the gender gap on adolescent smoking has been closed during the period. Decreased access to cigarettes has been associated with decreased smoking and the results show that better implementation of legislation is possible and should lead to further declines in the prevalence of smoking. The perception of the risks of smoking however has not increased, suggesting that targeted high-intensity tobacco control media campaigns may help and should be implemented.

The results also suggest that parents could contribute to further declines in smoking by enhancing monitoring of offspring. It is likely that adolescent smoking will reach the TFI target of 5% by 2025 given the rate of decline in this age group.

**Acknowledgements** K Babineau for planning and data collection of Irish ESPAD 2015 survey, Mark Morgan and Zubair Kabir for advice and support. The authors also acknowledge the ESPAD international researchers supported by EMCDDA contract CC.14.SDI.032 who compiled a common ESPAD Trend database (1995–2015).

**Contributors** LC had the idea for the study, organised the team, secured the funding, helped with data collection, analysis, data interpretation and writing. SL expanded the databases, did the final analyses and wrote a draft of paper. KT compiled the original datasets and did the initial analyses and initial draft writing. SK organised the data collection, helped with initial analysis and writing of all drafts.

**Funding** This work was supported by a grant made under an RFT for research services by the DOH for the ESPAD survey 2015; RCDHT Grant 178 supports SL.

**Competing interests** None declared.

**Patient consent** Not required.

**Ethics approval** DIT Dublin Ethics Committee.

**Provenance and peer review** Not commissioned; externally peer reviewed.

**Data sharing statement** The data were from the European School Survey Project on Alcohol and Other Drugs (ESPAD) and various official reports available from http://www.espad.org/reports-documents. With the 2003 data collection as a starting point, it was decided that all country datasets should be merged into a common database. After that also data from 2007 and 2011 are available in separate databases. Initially, these databases were stored and maintained by the Databank Manager Thoroddur Bjarnson. During the 2015 wave of ESPAD, the international database was compiled and standardised by CAN (Stockholm). Even though, since 2007, countries are obliged to deliver their national datasets to the database, there are—as stated in the database rules—no obligations to let other researchers use the national data without permission. In order to obtain a copy of a database, an application form has to be filled in and posted to the coordinators for further distribution to the ESPAD Application Committee. The composition of the committee as well as restrictions around the database and its use are described and explained in the ESPAD database rules (database rules for ESPAD

researchers and database rules for non-ESPAD researchers). When an application is approved, a contract is signed before a copy of the database is delivered. Approved applications are presented in a list, which also displays the deadline of the projects. ESPAD researchers are allowed to apply for the most recent database once the International ESPAD Report has been released. Non-ESPAD researchers are also allowed to work with ESPAD data. Access for non-ESPAD researchers is allowed after an embargo period determined by an assembly: ESPAD 2003 Database: accessible now. ESPAD 2007 Database: was accessible since 1 July 2013. ESPAD 2011 Database: was accessible since 1 July 2015. ESPAD 2015 Database: at present, it is only accessible to ESPAD researchers.

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
