## [Reviewer comments · BMJ Open]

ARTICLE DETAILS

TITLE (PROVISIONAL)	Decline of Adolescent Smoking in Ireland 1995-2015: Trend Analysis and Associated Factors
AUTHORS	Li, Shasha; Keogan, Sheila; Taylor, Keishia; Clancy, Luke

VERSION 1 – REVIEW

REVIEWER	Sofia B Ravara Faculty of Health Sciences, University of Beira Interior and CHCB University Hospital, Covilhã, Portugal.
REVIEW RETURNED	15-Sep-2017

GENERAL COMMENTS	General comment: This is a relevant paper that analyses the evolution of smoking behaviour among youths in Ireland, exploring the associations with diverse factors with a gender lens. Authors should include the following changes: Abstract: On the objectives, I suggest that authors would include the time frame of the study. Methods: Instead of “Logistic regressions were performed to examine...”, I suggest that authors would write “multivariate logistic regression was performed to examine...” On the independent variables of the MLR, instead of “socioeconomic status”, it should be indicated “perceived relative wealth”, as depicted in table 2. This variable is self-reported and based on students’ perception; it is not an objective SES calculation. Results: I suggest that authors would write “multivariate logistic regression shows that peer smoking ... instead of “regression results show that peer smoking” Conclusions I suggest that authors would include a statement related to the impact of comprehensive policies in closing the gender gap on youth smoking. Manuscript Introduction I suggest that authors would describe the package of strong tobacco control (TC) policies implemented in Ireland since 1995 showing the sequence of policies implemented overtime. In addition, details about enforcement and monitoring should be briefly explained. This information would be crucial to guide policy makers, especially those from countries with poor TC implementation, on the effectiveness and impact of strong TC policies in reducing youth smoking. To improve readability, I also suggest that authors would depict this in a figure showing the time frame, the type of TC policy and the smoking prevalence overtime. On page 6, lines 51-58, authors stated “We explored the relationship to smoking prevalence of gender,
--

	socioeconomic status, school performance, parental monitoring, peer smoking, perceived ease of access to cigarettes..” Instead of “socioeconomic status”, it should be indicated “perceived relative wealth”as depicted in table 2 . This variable is self-reported and based on students’ perception; it is not an objective SES calculation. Results On page10, lines 52-54, p value should be $p < 0.01$. The same change should be systematically done on table 3 and along the manuscript. On table 4, which depicts the MLR results ORs should include 95% CI intervals; the OR value for each reference should be 1 and not 0.00. In addition, the OR reference for year 2007 should be 1. I cannot understand the data on this last MLR analysis (regarding the 2007; 2011 and 2015 years) . May authors review and explain these data, please? Discussion On page 14, lines 13-20, while authors summarise the study findings, I suggest that authors would add the general trend that most factors impact more in girls than boys, specifying the 2 only exceptions. On page 15, lines 47-54, while authors discuss the trend on the perceived risks of cigarettes, I suggest that authors would also add that no prevention school-based programs were developed in Ireland during this period. I also suggest that authors while discussing the SES associations with smoking would consider that this variable was estimated from perceived family wealth and is not an objective measure. This may raise bias on the estimation of SES. Authors should discuss the limitations of cross-sectional studies and also the fact that factors associated with smoking were only available since 2007. Conclusions I suggest that authors would include a statement related to the impact of comprehensive policies in closing the gender gap on youth smoking.
--	---

REVIEWER	Hyeon Woo Yim Department of Preventive Medicine, College of Medicine, The Catholic University of Korea
REVIEW RETURNED	28-Sep-2017

GENERAL COMMENTS	This article has a number of problems that make it unacceptable for publication.  1. The decline of youth smoking has already been reported through ESPAD, so I think there is no novelty on this study results. 2. The authors have presented Tobacco control policies in Ireland, however, this study does not explain how they affected on the youth smoking rates. 3. The validity and reliability of mentioned the risk factors related to youth smoking were not presented in detail in the Method section and the operational definition of the variables in the manuscript remain unclear. 4. The contents between the results and the discussion were not linked each other. The discussion and conclusions are justified by the results. 5. The authors emphasized that 30-day smoking rates of both boys and girls are closed in 2015. Its public health implications did not mention in the manuscript.
---

REVIEWER	Kristiina Patja Pro Medico, Association for continuous medical education in Finland Helsinki, Finland Member of the scientific board for Tobacco-free Finland 2030
REVIEW RETURNED	05-Oct-2017

GENERAL COMMENTS	Review of BMJ open Kristiina Patja This is a very interesting and important trend analysis of youth smoking. Tobacco policies aim to influence three main processes: initiation, cessation and morbidity and mortality. This article describes policies in Ireland from 2007 to 2015 and also trend information from 1995-2005 in addition. Article provides important knowledge over effects of tobacco policies in tobacco initiation process. Some notions and requests to revise. Please excuse the language for non-native speaker, I try to be clear. First. PubMed term for youth is adolescence: please see https://www.ncbi.nlm.nih.gov/mesh/68000293 Abstract Page 2: What is ESPAD? It is not opened in the material and methods. Readers may not all be aware of this study acronym. Line 21: I prefer person over student as after school teens like be persons rather than teens or students. Line 24-28: Similar dependents asking similar factors together: family structure goes with SES, truancy is between family dependents. Conclusion lines 50-55: When did you have greatest decline= what was the most effective policy measure. It looks like that age limits in 2007 was a great change. It must have changed also the social norm about smoking that will influence parents and teachers. I would like to a conclusion rather than a statement over what happened. Page 5: Introductions Justification for all tobacco policies is the premature mortality and excess morbidity, but when writing on adolescence, this is not really necessary sentence. Start with policies than have been shown effective in previous research:  1) Access to tobacco products: advertising, number point of sales, packaging... 2) Display and visibility of tobacco products in environments: advertising, smoking bans... 3) Price of tobacco: package sizes, loose leaf tobacco e-cigarettes 4) Health education in general at school and special campaigns: health literacy and tobacco health harm awareness 5) Availability of cessation to young people: quitlines, school healthcare 6) Change in social norms by limiting adult tobacco use and tobacco use environments in the society
---

There are legislative initiations in this article to tackle these items, so why not point them and see which question items refer to these indirectly or directly. This introduction needs an aiming point to the results and discussion. First three paragraphs do not really add anything interesting in regarding the results. It would be better to concentrate on youth issues, not tackle the whole tobacco policy. There are literature reviews on all of the major influencing factors, even from world bank. E.g. Tob Control. 2014 Nov;23(e2):e89-97. doi: 10.1136/tobaccocontrol-2013-051110. Epub 2013 Sep 17. Impact of tobacco control interventions on socioeconomic inequalities in smoking: review of the evidence. Hill S1, Amos A2, Clifford D3, Platt S3.

Did you look at Cochrane publications over smoking initiation prevention and policies?
E.g. Cochrane Database Syst Rev. 2007 Jan 24;(1):CD004493. Family-based programmes for preventing smoking by children and adolescents. Thomas RE1, Baker P, Lorenzetti D.

Impact of tobacco advertising and promotion on increasing adolescent smoking behaviours
Chris Lovato , Allison Watts and Lindsay F Stead
Online Publication Date: October 2011

How was the tobacco price? You are missing the index prices here, that you obviously can get from national statistics. Real price index is usually shown also in OECD statistics. Introduction needs revision to match the objective of this article. It can also be shortened.

Material and methods:

Page 7: line 7: Open acronym ESPAD

Just to be sure for systematic bias: Is compulsory education in Ireland until 16, so full age-group is accessed at school?

Table 1. Please give sample sizes for 1995-2005 too. What is the response and non-response rates? Do they vary over the years?

Measures page 8:

I would start by defining what do you mean by smoking and non-smoking status. It needs to be first to tell. Did you ask other forms of tobacco products as by 2005 Swedish match started to market oral snuff and in many countries athlete boys started to use it first then later it spread to other boys too. How about e-Cigarette use? I have a funny feeling, that one factor in your decline between 2001 to 2015 is actually e-Cigarette. You must have some national datasets on this? You need to discuss this in this article. How about NRT products if they came into retail stores in 2014?

It would be reader friendly to think and open in text that you were thinking specific policy measure and also years they were put into force and then describing the questions used in evaluating these

hypothesis. Currently there is no logic in order of presenting quotations.

Statistical analysis

Page 9: Line 57: I do not understand this risk-question: what is the actual formulation of it? Is it a health risk defined in some disease or general risk of what?

Page 10: table 2 continues oddly in my version.

Page 10: line 21: small letters in variable, others in capital letters

Results:

Page 11: Table 3.

Is testing for gender difference really important? Trend analysis for 4-year change might be more interesting. In discussion, you do not discuss which policy measures were then important in producing this change and unless controlling for oral tobacco and e-Cigarette, this is a bit hasty.

What happened in 2003-2007? Were there some legislative actions that changed adolescence life in regarding tobacco use?

Predictors of youth smoking

There is a wide literature on tobacco use initiation risk factors. E.g. Am J Prev Med. 2016 Nov;51(5):767-778. doi: 10.1016/j.amepre.2016.04.003. Epub 2016 May 11.

Predictors of the Onset of Cigarette Smoking: A Systematic Review of Longitudinal Population-Based Studies in Youth. Wellman RJ1, Dugas EN2, Dutczak H3, O'Loughlin EK4, Datta GD3, Lauzon B3, O'Loughlin J5.

It would be good to read these and then look at some things that are different or surprising in Ireland. It is not really interesting to international readers with know-how on these issues to learn that it is same in Ireland than in most countries. If you have some specific interest in family structure, employment, recession or other external factors influencing the social control in Ireland, that could be interesting.

Page 12. Table 4: Odds ratios are usually shown before p-values.

Again in page 12, line 16: risk of smoking. In most articles this means that a person has this risk to be a smoker, but I assume this is not what you mean?

Discussion

Page 14:

It is fine to collect main findings in to the first paragraph, but then I am missing the communication between your findings of interdependencies between policies and prevalences and then discussion over the reasons that will lead to these assumptions. Then you will find some research to back your thoughts.

I would like to see discussion in order of policies that have been shown effective in previous research: Access to tobacco products, display and visibility of tobacco products in environments, price of tobacco, health education in general at school and special campaigns: health literacy and tobacco health harm awareness,

	availability of cessation and then change in social norms. In this format discussion is mainly a repetition of results. E-cigarette needs to be taken into consideration in discussion for low frequencies. In other countries like Us or in Finland, there is a declining trend on smoked cigarettes with increase of e-Cigarettes. Prevalence of nicotine users among adolescence is actually increasing in some populations. This article consists of the blocks, but they need to be organized into interesting and logic story. Please do find time and energy to do so.
--	---

REVIEWER	Arja Rimpelä University of Tampere, Faculty of Social Sciences, Health Sciences, Finland
REVIEW RETURNED	14-Oct-2017

GENERAL COMMENTS	Declining trends in smoking are always of interest as well as factors explaining them. The problem of the paper is that it is not using data to analyse the trends and factors behind them. The reader is expecting to see how different factors have contributed to the decline or at least if the decline has been similar in all groups. This analysis is lacking. Now data from 2007 and 2015 are pooled. The factors included in the analysis, do not bring anything new to the literature. That is why the cross-sectional analysis is not interesting for the reader. The effect of survey year is of interest and this should be analysed e.g. by the interaction term with each explaining factor. The main question of the study should be what factors between 2007 and 2015 have contributed to the decline. Are there any changes in the risk factors for smoking? The research questions should be reformulated. The list of factors selected for the analysis is not well-argued. The collection of variables belong to many different dimensions and it would be easier to read if the variables would be categorized in the tables. Are all these variables really needed? Why school performance is not included here? Was it not asked? It is one of the most powerful predictors and there is evidence that smoking is declining less among those with poor school performance. The table does not show bivariate connections between smoking and the variables. These are needed. If I understand, the table has all the variables put in the same model. (The title of the table is not acceptable) There surely are correlations between many variables why multivariate models may give strange results and lose some important connections. The authors refer to legislation e.g. health warnings. The timing of surveys and the questions used do not confirm that the legislative change did not have an effect. According to health behaviour theories, the effect on behavior is not the first place to find an effect. And the other direction would be more interesting: did the smokers see the risks as often as non-smokers and did this change. I would keep the discussion on legislative changes on a general level. The changed societal atmosphere made it possible to pass these legislative changes and vice versa why causal connections with a simple analysis should not be drawn. Coming back to the explaining variables. What is the role of each of them? It is self-evident that if youth smoking goes down, there are less smoking friends, too. Or if the authors want to show that smoking friends are not as important/more important risk factor now
--

	than earlier, then they should analyse this. The role of each variable should be considered. The data are purely described, the 1995 data is not presented in the table. It is unclear what the authors mean with missing data from 1995. Does this mean that Ireland has lost its data? I wonder if it is worth of dropping the 1995 data in the methods but using some already published papers to show changes in the gender difference; the picture can still be drawn, no statistical analyses are needed. This would make the paper more coherent and also the title of the paper could be 2007-2015 which would better describe the content. If the data from 1995 is available, there are some questions on socio-economic differences which might give comparison with the new data and show if smoking has changed in some groups. In its present form, the title gives a false impression on the content because it makes the reader to expect full analyses of factors having effect on smoking. Finally, Strength and limitations are not either strengths or limitations. E.g. there seems to be better data to describe changes in Ireland, namely in HBSC. Multivariate model that controls the confounders could be a valid argument if a single causal relationship would be studied but this is not the case here. Strengths could be e.g. that the data sets in different years were comparable in data collection, questions etc. Altogether, I suggest authors to reconsider they research questions, analysis and the scope of the study.
--	--

VERSION 1 – AUTHOR RESPONSE

Reviewers' Comments to Author:

Reviewer: 1

Reviewer Name: Sofia B Ravara

Institution and Country: Faculty of Health Sciences, University of Beira Interior and CHCB University Hospital, Covilhã, Portugal.

Competing Interests: None declared.

General comment: This is a relevant paper that analyses the evolution of smoking behaviour among youths in Ireland, exploring the associations with diverse factors with a gender lens. Authors should include the following changes:

Abstract:

On the objectives, I suggest that authors would include the time frame of the study.

The time frame of the study is included as suggested. We've also mentioned the different time frames used in examining the gap (between 1995 and 2015) and in assessing the factors associated with smoking behaviours (between 2007 and 2015).

Methods:

Instead of "Logistic regressions were performed to examine...", I suggest that authors would write "multivariate logistic regression was performed to examine..."

Changed to "multivariate logistic regression" in Methods and other corresponding places.

On the independent variables of the MLR, instead of "socioeconomic status", it should be indicated "perceived relative wealth", as depicted in table 2. This variable is self-reported and based on students' perception; it is not an objective SES calculation.

Changed to "perceived relative wealth" in Methods and other corresponding places.

Results: I suggest that authors would write "multivariate logistic regression shows that peer smoking ... instead of "regression results show that peer smoking"

Changed to "multivariate logistic regression...".

Conclusions

I suggest that authors would include a statement related to the impact of comprehensive policies in closing the gender gap on youth smoking.

Added in Conclusion that “the gender gap on adolescent smoking has been closed during the period”.

Manuscript
Introduction

I suggest that authors would describe the package of strong tobacco control (TC) policies implemented in Ireland since 1995 showing the sequence of policies implemented overtime. In addition, details about enforcement and monitoring should be briefly explained. This information would be crucial to guide policy makers, especially those from countries with poor TC implementation, on the effectiveness and impact of strong TC policies in reducing youth smoking. To improve readability, I also suggest that authors would depict this in a figure showing the time frame, the type of TC policy and the smoking prevalence overtime.

Actually, in the Appendix of the original manuscript, there was a figure showing the timeline of all types of tobacco control policies implemented in Ireland since 1995. In addition, we have rewritten the Introduction to cover more tobacco control policies implemented in Ireland since 1995.

On page 6, lines 51-58, authors stated “We explored the relationship to smoking prevalence of gender, socioeconomic status, school performance, parental monitoring, peer smoking, perceived ease of access to cigarettes..” Instead of “socioeconomic status”, it should be indicated “perceived relative wealth” as depicted in table 2. This variable is self-reported and based on students’ perception; it is not an objective SES calculation.

Changed to “perceived relative wealth” here and other corresponding places.

Results

On page 10, lines 52-54, p value should be $p < 0.01$. The same change should be systematically done on table 3 and along the manuscript.

We have systematically changed the display of p value throughout the manuscript including tables according to the rule accepted by most journals as following: If p value is smaller than 0.001, report it as < 0.001 ; if p value is smaller than 0.01, report it to the nearest thousandth; if p value is smaller than 0.99, report it to the nearest hundredth; if p value is larger than 0.99, report it as > 0.99 .

On table 4, which depicts the MLR results ORs should include 95% CI intervals; the OR value for each reference should be 1 and not 0.00. In addition, the OR reference for year 2007 should be 1. I cannot understand the data on this last MLR analysis (regarding the 2007; 2011 and 2015 years). May authors review and explain these data, please?

Actually, I was reporting the p-value instead of OR for some reference categories as 0.00. Anyway I have changed the table by reporting OR and 95% CI and making the variables or categories bold if they are significant at 0.05 level.

Essentially, data used for the multivariable logistic regression are based on the last three waves of survey years, i.e., 2007, 2011 and 2015. We did not include the data of 1999 and 2003, as we only had aggregated data, raw data are not available. We did not include the 1995 data, as most of the variables we are interested were not asked in the questionnaire of 1995.

The dependent variable is whether a respondent is a current smoker or not. The independent variables are the measures shown in Table 4. The definitions of the measures used in the regressions are explained in “Predictors of youth smoking”.

Discussion

On page 14, lines 13-20, while authors summarise the study findings, I suggest that authors would add the general trend that most factors impact more in girls than boys, specifying the 2 only exceptions.

Actually, although some ORs looked greater for girls than boys. They are not significantly different. I have run a separate regression including interaction of gender and all other measures, to capture if the impacts differed between boys and girls. It turns out that only two interaction terms are significant. They are “parents know Saturday nights” and “One parent”. In particular, boys are more likely to smoke if their parent usually don’t know where they spend Saturday nights than girls. Being in one-parent family is related to higher risk of smoking for girls, not for boys. For other variables, the impacts are not significantly different between boys and girls. Therefore, I could not add the general trend that most factors impact more in girls than boys.

On page 15, lines 47-54, while authors discuss the trend on the perceived risks of cigarettes, I suggest that authors would also add that no prevention school-based programs were developed in Ireland during this period.

We have added as suggested that “no school-based programs were developed in Ireland aiming to raise the awareness of risks associated with smoking during this period”.

I also suggest that authors while discussing the SES associations with smoking would consider that this variable was estimated from perceived family wealth and is not an objective measure. This may raise bias on the estimation of SES.

We have added discussion on the limitation of the variable we used to capture SES as follows. "There are some explanations for the absence of effect. First, although we meant to capture socioeconomic status by perceived family wealth compared to other student, the measure is not objective, which might raise bias on the estimation of socioeconomic effect. For example, if most of a student's friends are from very wealthy families, despite of the student's real family wealth, the student might feel he is less well off than them, which will make the measure far from the true socioeconomic status." Authors should discuss the limitations of cross-sectional studies and also the fact that factors associated with smoking were only available since 2007.

We have added the limitation of the cross-sectional studies in Discussion, "...However, the direction of causality is not necessarily clear due to the limitation of cross-sectional study...."

We've also made it clear that the timeframe used to the regression analysis is since 2007 due to data limitation.

Conclusions

I suggest that authors would include a statement related to the impact of comprehensive policies in closing the gender gap on youth smoking.
Added as suggested.

Reviewer: 2

Reviewer Name: Hyeon Woo Yim

Institution and Country: Department of Preventive Medicine, College of Medicine, The Catholic University of Korea
Competing Interests: None declared

This article has a number of problems that make it unacceptable for publication.

1 The decline of youth smoking has already been reported through ESPAD, so I think there is no novelty on this study results.

We are the Irish PI of ESPAD and agree that the prevalence of youth smoking has fallen in most European countries. But it is very difficult to get a good idea of what is going on. The prevalence now in ESPAD countries varies from 5% to 37% in girls and 5% to 35% in boys. Mostly the prevalence is greater in boys and the gender gap is decreasing i.e. boys have a sharper decline than girls. It is the opposite in Ireland with a sharper decline in girls as female youth smoking was dramatically higher than male smoking in the 90's and the gap has been eliminated. And although there is a general decline in ESPAD countries there is no evidence of convergence in the different countries or geographic regions. The introduction of increased Tobacco Control (TC) measures is also unequal and it has proved impossible in ESPAD to-date to test the effects of different interventions in different countries. We agree this is very desirable and are trying to look at this in Ireland in other ways. There were no questions in ESPAD however about price, point of sale bans, youth access or advertising etc. However what we did here is to look at the factors which were surveyed in all ESPAD countries and look for those factors which might be important at the individual level and could be considered possibly as proxies for national level TC interventions. We did not think it would be valid to try to directly relate these TC interventions to the individual results we surveyed. The hope was that there would be observations from Irish data to suggest what might work more generally in other countries. Nevertheless we have expanded the discussion re possible relationship of our findings to various TC changes.

1. The authors have presented Tobacco control policies in Ireland, however, this study does not explain how they affected on the youth smoking rates.

That is true but as discussed above it is difficult to examine this issue using ESPAD data with such complex interventions and interactions especially as ESPAD is held only every 4 years and there were no questions on these interventions but we agree it would be important.

As discussed above we have explored this relationship further in the revised paper and hope to study these possible outcomes using other approaches.

2. The validity and reliability of mentioned the risk factors related to youth smoking were not presented in detail in the Method section and the operational definition of the variables in the manuscript remain unclear.

We have tried to clarify these matters in conjunction with the suggestions from the other referees.

3. The contents between the results and the discussion were not linked each other. The discussion and conclusions are justified by the results.

We have tried to make the relationships clearer and assume the criticism is that "The discussion and conclusions are not justified by the results." We have therefore expanded this discussion to try to clarify this relationship.

4. The authors emphasized that 30-day smoking rates of both boys and girls are closed in 2015. Its public health implications did not mention in the manuscript.

We were perhaps concentrating on the fact that the gap had been eliminated rather than the P.H. consequences as Ireland has agreed a plan to be tobacco free by 2025 and the elimination of the gender differential is important in that regard. We have commented on some possible implications now.

Reviewer: 3

Reviewer Name: Kristiina Patja

Institution and Country: Pro Medico, Association for continuous medical education in Finland, Helsinki, Finland; Member of the scientific board for Tobacco-free Finland 2030 Competing Interests: None declared

This is a very interesting and important trend analysis of youth smoking. Tobacco policies aim to influence three main processes: initiation, cessation and morbidity and mortality. This article describes policies in Ireland from 2007 to 2015 and also trend information from 1995-2005 in addition. Article provides important knowledge over effects of tobacco policies in tobacco initiation process.

Some notions and requests to revise. Please excuse the language for non-native speaker, I try to be clear.

First. PubMed term for youth is adolescence: please see <https://www.ncbi.nlm.nih.gov/mesh/68000293>

We have changed the term from "youth" to "adolescent".

Abstract Page 2:

What is ESPAD? It is not opened in the material and methods. Readers may not all be aware of this study acronym.

The full name is stated now in the Abstract. The survey is also explained in detail in the manuscript

Line 21: I prefer person over student as after school teens like be persons rather than teens or students.

As the survey was based on 14-16 school students, we think it is probably clearer to still refer them as students for this study.

Line 24-28: Similar dependents asking similar factors together: family structure goes with SES, truancy is between family dependents.

They are now reordered as follows: gender, survey years, perceived ease of access to cigarettes, perceived risk of smoking, perceived relative wealth, parental monitoring, maternal relationship, family structure, truancy and peer smoking.

Conclusion lines 50-55: When did you have greatest decline= what was the most effective policy measure. It looks like that age limits in 2007 was a great change. It must have changed also the social norm about smoking that will influence parents and teachers. I would like to a conclusion rather than a statement over what happened.

It is not easy to conclude when we had greatest decline, the policies in the period were most effective, due to lagging effect or synergistic effect. The age change became law in 2002 but was not fully implemented until 2007. Whereas the Smokefree Law was implemented in 2004. Besides, other policies not specifically targeting adolescents could also contribute to the reduction in prevalence. For example, in 2007, a ban was introduced on packets containing less than 20 cigarettes. This policy could make a single packet of cigarettes more expensive, especially for adolescents who are most financially constrained. Therefore it is not easy to attribute the reduction in prevalence to one single policy.

Page 5: Introductions

Justification for all tobacco policies is the premature mortality and excess morbidity, but when writing on adolescence, this is not really necessary sentence. Start with policies than have been shown effective in previous research:

- 1) Access to tobacco products: advertising, number point of sales, packaging...
- 2) Display and visibility of tobacco products in environments: advertising, smoking bans...
- 3) Price of tobacco: package sizes, loose leaf tobacco e-cigarettes
- 4) Health education in general at school and special campaigns: health literacy and tobacco health harm awareness
- 5) Availability of cessation to young people: quitlines, school healthcare
- 6) Change in social norms by limiting adult tobacco use and tobacco use environments in the society

There are legislative initiations in this article to tackle these items, so why not point them and see which question items refer to these indirectly or directly. This introduction needs an aiming point to the results and discussion. First three paragraphs do not really add anything interesting in regarding the results. It would be better to concentrate on youth issues, not tackle the whole tobacco policy.

Introduction was rewritten, focusing on the importance of preventing adolescent smoking, what has been done in Ireland in terms of tobacco control policies and what the existing literature has found regarding to factors associated with adolescent smoking behaviours.

There are literature reviews on all of the major influencing factors, even from world bank. E.g. Tob Control. 2014 Nov;23(e2):e89-97. doi: 10.1136/tobaccocontrol-2013-051110. Epub 2013 Sep 17. Impact of tobacco control interventions on socioeconomic inequalities in smoking: review of the evidence.

Hill S1, Amos A2, Clifford D3, Platt S3.

Did you look at Cochrane publications over smoking initiation prevention and policies?

E.g. Cochrane Database Syst Rev. 2007 Jan 24;(1):CD004493.

Family-based programmes for preventing smoking by children and adolescents.

Thomas RE1, Baker P, Lorenzetti D.

Impact of tobacco advertising and promotion on increasing adolescent smoking behaviours Chris Lovato , Allison Watts and Lindsay F Stead Online Publication Date: October 2011

The last paragraph of Introduction was a review of what existing literature has found regarding to factors related to adolescent smoking, which also justified why we have chosen the measures in our regression analysis.

How was the tobacco price? You are missing the index prices here, that you obviously can get from national statistics. Real price index is usually shown also in OECD statistics.

Another new Appendix was added to show the real price change of a packet of 20 cigarettes between 1995 and 2015. It is also now mentioned in the 2nd paragraph of Introduction.

Introduction needs revision to match the objective of this article. It can also be shortened.

Introduction has been rewritten and shortened, focusing on the importance of preventing adolescent smoking, what has been done in Ireland in terms of tobacco control policies and what the existing literature has found regarding to factors associated with adolescent smoking behaviours.

Material and methods:

Page 7: line 7: Open acronym ESPAD

It has now been written in full.

Just to be sure for systematic bias: Is compulsory education in Ireland until 16, so full age-group is accessed at school?

Yes, education is compulsory for children in Ireland from the age of 6 to 16 or until students has completed three years of second-level education.

Table 1. Please give sample sizes for 1995-2005 too. What is the response and non-response rates? Do they vary over the years?

Table 1 now includes sample size, gender ratio, and response rate for all 6 waves of data.

Measures page 8:

I would start by defining what do you mean by smoking and non-smoking status. It needs to be first to tell.

The definition of smokers are updated in the beginning of Measures as follows. "Respondents were asked how frequently they had smoked in the last 30 days, with answers ranging from "not at all" to "more than 20 cigarettes per day". Those who answered 'not at all' are non-smokers and those who had smoked at least once in the last 30 days are smokers. Current or 30-day smoking prevalence rate is the proportion of smokers."

Did you ask other forms of tobacco products as by 2005 Swedish match started to market oral snuff and in many countries athlete boys started to use it first then later it spread to other boys too. How about e-Cigarette use? I have a funny feeling, that one factor in your decline between 2001 to 2015 is actually e-Cigarette. You must have some national datasets on this? You need to discuss this in this article. How about NRT products if they came into retail stores in 2014?

Actually the ESPAD survey did not ask about other forms of tobacco products. Except in 2007 survey, it asked about snuff, but it was just for that year, as snus is not available on the Irish market. NRT is only available in Pharmacies is not for sale to under 18s except with doctor's prescription

As to e-cig, it was only asked in the 2015 survey and we think only in Ireland as we were allowed add a small number of questions. It is not reported in the Official ESPAD 2015 report except in the local Irish report. Ecigs only became available on Irish market in 2011.

Therefore, we are not able to get the trends in other forms of tobacco products in ESPAD study such as snuff and e-cig and compare them with the cigarette smoking found in our study.

However, we are involved in two projects on e-cigs. One was a cross sectional study of prevalence and associated factors of e-cig usage among Irish youths, which was conducted in 2014. The study has been published (See Babineau, K., Taylor, K., & Clancy, L. (2015). Electronic cigarette use among Irish youth: a cross sectional study of prevalence and associated factors. PLoS One, 10(5), e0126419.). Another recent study on e-cigs is a working paper as part of the Project SILNE-R HORIZON 2020, which is to assess how recent strategies and programs to prevent youth smoking have been implemented at national, municipal and school levels and how they have influenced smoking behaviour of 16 year old adolescents in seven European countries. We have obtained the data and are working on it.

We do not know if ecigs are delaying the decline but have no evidence that it increased the rate of decline but are worried about the future effects. We feel it was not relevant in the noted decline but we will continue to observe the possible effects.

It would be reader friendly to think and open in text that you were thinking specific policy measure and also years they were put into force and then describing the questions used in evaluating these hypothesis. Currently there is no logic in order of presenting quotations.

The Introduction section was rewritten, focusing on the importance of preventing adolescent smoking, what has been done in Ireland in terms of tobacco control policies and what the existing literature has found regarding to factors associated with adolescent smoking behaviours.

Statistical analysis

Page 9: Line 57: I do not understand this risk-question: what is the actual formulation of it? Is it a health risk defined in sme disease or general risk of what?

The risk measurement was explained in Methods-Measure section as follows. "Students were asked how difficult it would be to get cigarettes if they wanted them and to what extent people risk harming themselves (physically or in other ways) if they smoke occasionally. The majority of students thought it would be easy to get cigarettes and about half of the students perceived a moderate/great risk from smoking occasionally."

Page 10: table 2 continues oddly in my version.

It should be ok now.

Page 10: line21: small letters in variable, others in capital letters

Changed to capital letters.

Results:

Page 11: Table 3.

Is testing for gender difference really important? Trend analysis for 4-year change might be more interesting.

In other countries in ESPAD study, mostly the prevalence is greater in boys and the gender gap is decreasing i.e. boys have a sharper decline than girls. It is the opposite in Ireland with a sharper decline in girls as female youth smoking was dramatically higher than male smoking in the 90's and the gap has been eliminated. Although it is not easy to pinpoint what caused the closure of the gap given the frequency of the survey (i.e. every 4 years) and the availability of the data, we feel it is still an interesting and maybe important topic.

In discussion, you do not discuss which policy measures were then important in producing this change and unless controlling for oral tobacco and e-Cigarette, this is a bit hasty.

In Discussion, we have discussed the importance of some policies that might lead to the changes in the factors that found to be significant in the regressions. For example, perceived ease of access to cigarettes is found in the regression results to be a significant factor related to adolescent smoking,

and the measure itself has improved between 2007 and 2015. Therefore we discussed what policies might contribute to the improvement of the measure as follows in Discussion. "Perceived ease of access to cigarettes decreased between 2007 and 2015. Students claiming that it was difficult to get cigarettes increased from 12% in 2007 to 28% in 2015. Several policies introduced during this period might contribute to the increase in difficulty accessing to cigarettes. In particular, the implementation of age limit law under Public Health (Tobacco) Act 2002 has officially started since April 2007. It is an offence to sell cigarettes or other tobacco products to persons aged under 18 years. In addition, since 2009 retailers are required to register with the National Tobacco Control Office...."

As stated above, the ESPAD study does not have enough information on other oral tobacco production and e-cigarette.

What happened in 2003-2007? Were there some legislative actions that changed adolescence life in regarding tobacco use?

Between 2003 and 2007, one of the most important policies might be the smoke-free law, which applied to all worksites, including bars and restaurants. However, it was not specifically targeting adolescents.

Predictors of youth smoking

There is a wide literature on tobacco use initiation risk factors. E.g. Am J Prev Med. 2016 Nov;51(5):767-778. doi: 10.1016/j.amepre.2016.04.003. Epub 2016 May 11.

Predictors of the Onset of Cigarette Smoking: A Systematic Review of Longitudinal Population-Based Studies in Youth. Wellman RJ1, Dugas EN2, Dutczak H3, O'Loughlin EK4, Datta GD3, Lauzon B3, O'Loughlin J5.

It would be good to read these and then look at some things that are different or surprising in Ireland. It is not really interesting to international readers with know-how on these issues to learn that it is same in Ireland than in most countries. If you have some specific interest in family structure, employment, recession or other external factors influencing the social control in Ireland, that could be interesting.

There is a wealth of evidence to link adolescent smoking and low socioeconomic status, but this study has found that perceived lower relative wealth was not linked to an increase likelihood of smoking, which was discussed in Discussion section. Unfortunately, the choices of the measures included in the regression analysis is limited by the survey questions.

Page 12. Table 4: Odds ratios are usually shown before p-values. Again in page 12, line 16: risk of smoking. In most articles this means that a person has this risk to be a smoker, but I assume this is not what you mean?

Data in Table 4 are reordered and presented in a more concise and precise way by including 95% CI of OR and bolding the significant factors at level 0.05.

The risk measure is now displayed as "perceived risk of smoking" and explained in detail in Measures section. It measures to what extent people risk harming themselves (physically or in other ways) if they smoke occasionally.

Discussion

Page 14:

It is fine to collect main findings in to the first paragraph, but then I am missing the communication between your findings of interdependencies between policies and prevalences and then discussion over the reasons that will lead to these assumptions. Then you will find some research to back your thoughts.

The logic in Discussion is as follows: it talks about the measures in the multivariable logistic regression.

If those measures are found significant in the regression, we will further discuss the changes in the measures between 2007 and 2015 and what tobacco control policies might lead to the changes. For example, perceived ease of access to cigarettes is found in the regression results to be a significant factor related to adolescent smoking. Therefore we talk about that the measure itself has improved between 2007 and 2015 according to Table 2. Then we discuss what policies might contribute to the

improvement of the measure, such as the implementation of age limit law in 2007, requiring retailers to register with the National Tobacco Control Office, etc.

If those measures are insignificant in the regression, we will further discuss if they are significant in existing literature and why it is insignificant in our study. For example, there is a wealth of evidence to link adolescent smoking and low socioeconomic status, but this study has found that perceived lower relative wealth was not linked to an increase likelihood of smoking. Therefore, we discussed the reasons behind it.

In addition, we also discussed the limitation of the study.

I would like to see discussion in order of policies that have been shown effective in previous research: Access to tobacco products, display and visibility of tobacco products in environments, price of tobacco, health education in general at school and special campaigns: health literacy and tobacco health harm awareness, availability of cessation and then change in social norms. In this format discussion is mainly a repetition of results. E-cigarette needs to be taken into consideration in discussion for low frequencies. In other countries like Us or in Finland, there is a declining trend on smoked cigarettes with increase of e-Cigarettes. Prevalence of nicotine users among adolescence is actually increasing in some populations.

The order and logic of the Discussion section is stated in the response above.

ESPAD does not have trend data on e-cig as it just recently included it in 2015 survey and only in Ireland nor is there any other database for ecigs in school students in Ireland and see above other than the survey TFRI published.

This article consists of the blocks, but they need to be organized into interesting and logic story. Please do find time and energy to do so.

The paper has been updated and displayed in a more logic way as suggested.

Reviewer: 4

Reviewer Name: Arja Rimpelä

Institution and Country: University of Tampere, Faculty of Social Sciences, Health Sciences, Finland

Competing Interests: None declared

Declining trends in smoking are always of interest as well as factors explaining them. The problem of the paper is that it is not using data to analyse the trends and factors behind them.

The reader is expecting to see how different factors have contributed to the decline or at least if the decline has been similar in all groups. This analysis is lacking. Now data from 2007 and 2015 are pooled. The factors included in the analysis, do not bring anything new to the literature. That is why the cross-sectional analysis is not interesting for the reader. The effect of survey year is of interest and this should be analysed e.g. by the interaction term with each explaining factor. The main question of the study should be what factors between 2007 and 2015 have contributed to the decline. Are there any changes in the risk factors for smoking? The research questions should be reformulated.

The paper uses the aggregated data between 1995 and 2015 to show the trends in smoking prevalence among the 15-16 year old students. It also uses the cross-sectional data between 2007 and 2015 to show the factors associated with youth smoking behaviours.

We have attempted to see if survey years are of interest by including interaction terms of survey years and all other measures. However, the interaction terms are not significant.

We have discussed the changes in the risk factors for smoking. For example, from the multivariate regression, perceived ease of access to cigarettes is found in the regression results to be a significant factor related to adolescent smoking, and the measure itself has improved between 2007 and 2015. Therefore we discussed what policies might contribute to the improvement of the measure in Discussion.

The list of factors selected for the analysis is not well-argued. The collection of variables belong to many different dimensions and it would be easier to read if the variables would be categorized in the tables. Are all these variables really needed? Why school performance is not included here? Was

it not asked? It is one of the most powerful predictors and there is evidence that smoking is declining less among those with poor school performance.

In the last paragraph of Introduction section, we have argued why those factors are included by linking to other literature. In addition, in the Measures section, we have shown in detail how the factors are measured.

We have reordered the factors. For example, the family related factors are placed together. The selection of the factors are based on the findings of existing literature and the availability of the datasets.

School performance is not included here because the questions asked about school performance are different across the years, which makes the variable not comparable. Instead, we have included truancy as recent studies show that poor grades is a significant predictor for truancy and truancy is superior measure than grade point. See For example "Who's Skipping School: Characteristics of Truants in 8th and 10th Grade" by Henry, Kimberly, *The Journal of School Health*; Kent Vol. 77, Iss. 1, (Jan 2007): 29-35. and "Truancy, Grade Point Average, and Sexual Activity: A Meta-Analysis of Risk Indicators for Youth Substance Use" by Hallfors, D., Vevea, J. L., Iritani, B., Cho, H., Khatapoush, S. and Saxe, L. (2002) *Journal of School Health*, 72: 205–211.

The table does not show bivariate connections between smoking and the variables. These are needed. If I understand, the table has all the variables put in the same model. (The title of the table is not acceptable) There surely are correlations between many variables why multivariate models may give strange results and lose some important connections.

Yes, we used multivariable logistic regression to assess the significance of the factors. We are afraid that bivariate connection between smoking and the variables won't take into account the impact of other factors.

The authors refer to legislation e.g. health warnings. The timing of surveys and the questions used do not confirm that the legislative change did not have an effect. According to health behaviour theories, the effect on behavior is not the first place to find an effect. And the other direction would be more interesting: did the smokers see the risks as often as non-smokers and did this change. I would keep the discussion on legislative changes on a general level. The changed societal atmosphere made it possible to pass these legislative changes and vice versa why causal connections with a simple analysis should not be drawn.

Coming back to the explaining variables. What is the role of each of them? It is self-evident that if youth smoking goes down, there are less smoking friends, too. Or if the authors want to show that smoking friends are not as important/more important risk factor now than earlier, then they should analyse this. The role of each variable should be considered.

The selection of the variables was based on the findings of existing studies and the availability of the data in ESPAD surveys. In Introduction section, we talked about the roles of the some of the variables in the existing literature.

In Discussion, we also talked about the policies that might affect the variables.

The data are purely described, the 1995 data is not presented in the table. It is unclear what the authors mean with missing data from 1995. Does this mean that Ireland has lost its data? I wonder if it is worth of dropping the 1995 data in the methods but using some already published papers to show changes in the gender difference; the picture can still be drawn, no statistical analyses are needed. This would make the paper more coherent and also the title of the paper could be 2007-2015 which would better describe the content. If the data from 1995 is available, there are some questions on socio-economic differences which might give comparison with the new data and show if smoking has changed in some groups. In its present form, the title gives a false impression on the content because it makes the reader to expect full analyses of factors having effect on smoking.

Now Table 1 has included sample statistics of all years. It is now also clearly explained in the paper what happened to the data of 1995, 1999 and 2003 as follows. "The 1999 and 2003 survey raw datasets are unavailable and the 1995 dataset did not include most of the measures used in the study." That is why in the multivariable logistic regression, only data from 2007 to 2015 are included. However, we

are able to get aggregate prevalence data for year 1999 and 2003. Therefore, when looking into the trend of smoking prevalence, all years are included.

Finally, Strength and limitations are not either strengths or limitations. E.g. there seems to be better data to describe changes in Ireland, namely in HBSC. Multivariate model that controls the confounders could be a valid argument if a single causal relationship would be studied but this is not the case here. Strengths could be e.g. that the data sets in different years were comparable in data collection, questions etc.

We did not use HBSC data in Ireland for several reasons.

1. HBSC data in Ireland is from 1998 to 2014, with only 5 waves, while ESPAD Ireland is from 1995 to 2015, with 6 waves.

2. HBSC is a general survey gaining insight into young people's well-being, health behaviours and their social context, instead of focusing on alcohol, smoking and other substance uses like ESPAD. Therefore HBSC does not provide enough measures/variables related to smoking and the factors. In particular, it only has 3 to 4 questions relating to smoking. It does not include questions on perceived difficulty in accessing to cigarettes and perceived health risks of smoking cigarettes. In addition, it does not have questions on family monitoring and peer smoking.

3. The sample from HBSC covering the similar age group as ESPAD is smaller than the one of ESPAD. The sample size of ESPAD is about 2,000 on average, while HBSC sample size ranges from 919 to 1,695.

Therefore, ESPAD seems to be the best data to tracking youth smoking behaviours in this age group.

Strengths like comparability in data is added as suggested.

Altogether, I suggest authors to reconsider their research questions, analysis and the scope of the study.

The objective of this study is to present the trend of youth smoking between 1995 and 2015, and to assess the factors associated with youth smoking behaviours based on detailed data between 2007 and 2015 in ESPAD. Other interesting research questions like what policies helped reduce prevalence in some period is on our agenda but is difficult to get at with present data sets.

VERSION 2 – REVIEW

REVIEWER	Kristiina Patja Pro Medico, Association for continuous professional development in Finland
REVIEW RETURNED	23-Nov-2017

GENERAL COMMENTS	This article was under my review in 5th October and my main concern was then authors should focus on policies and processes shown effective in previous research:  1) Access to tobacco products: advertising, number point of sales, packaging... 2) Display and visibility of tobacco products in environments: advertising, smoking bans... 3) Price of tobacco: package sizes, loose leaf tobacco e-cigarettes 4) Health education in general at school and special campaigns: health literacy and tobacco health harm awareness 5) Availability of cessation to young people: quitlines, school healthcare 6) Change in social norms by limiting adult tobacco use and tobacco use environments in the society I am really happy to see that this article has changed into an
--

	interesting piece of research for international audience such as me. You have demonstrated that legislative measures implemented will have an impact on access and visibility, which leads to decline in use. It also seems to have an impact on the social climate among adults, which could be seen in parenting practices regarding smoking. This then makes a positive cycle for non-tobacco use being a norm. I can understand that my previous comments could have been seen harsh, but after reading your work, perhaps they were somewhat relevant. You have interesting findings:  • You have been able to narrow the gender gap, which is great. No real reason perhaps can be detected. • Girls seem to be more vulnerable to poor relationships to their parents? • Parent do matter: you see in negative way, but this is a good message for parents in despair with adolescence: my controlling and caring matters. A good message. • Single parenting is not a bad thing, it is all about the quality of parenting • Change in prevalence changes social environments: page 8 line 43: when tobacco use vanishes, clues decrease and social acceptance disappears. Then your results change: this is obvious, not methodological problem. In the introduction I still miss the prevalence on e-Cigarette use as a background, oral snuff? Since this is a next era of battle against nicotine, I deserves a mentioning. E.g. http://www.tri.ie/uploads/5/2/7/3/52736649/shahsa_ectoh_e-cig.pdf Some hopes: I table titles they need to have more information. In e-publications we tend to use them sometimes separately and title needs to contain all relevant information to be understood. It is also your advantage to be acknowledged. I think I said something last time as well about this. You are discussing economical issues in discussion, is it seen in year 2011 perceived family relative wealth and then in 2015 less well off? In Finland recession left poverty to some families permanently. It is not a issue here, just wondering. Low SES does not mean high risk if parents get support. Conclusion could be more concrete: What did we do successfully? What we learned? What we learned that need to improve in legislation, in environments like schools, in social climate change like support for parents and then finally perhaps the gender specific programmes for boys and girls separately? You have them in your discussion.
--	---

REVIEWER	Sofia Ravara Faculty of Health Sciences, University of Beira Interior and CHCB University Hospital, Covilhã, Portugal.
REVIEW RETURNED	23-Nov-2017

GENERAL COMMENTS	Authors have adressed reviewers' concerns and suggestions. I have no further comments to add.
---

REVIEWER	Arja Rimpelä University of Tampere, Finland
REVIEW RETURNED	11-Dec-2017

GENERAL COMMENTS	Two minor things:  - The authors should mention in the end of the introduction page 6 row 27 the years when this aim was measured (2007-2015). Now the reader understands that the risk factors are measured for the entire period. - The first section of Discussion page 14, row 25 the authors wrote "...are associated with increased adolescent smoking". The word "increased" should be dropped because the paper deals with a decrease in smoking and it has not been studied that these factors increased smoking during the study period.
---

REVIEWER	Hyeon Woo Yim Department of Preventive Medicine, College of Medicine, Catholic University of Korea, South Korea
REVIEW RETURNED	20-Dec-2017

GENERAL COMMENTS	 1. It is unclear what the authors want to say through this paper. Is this about whether "trend of youth smoking between 1995 and 2015 in Ireland" or "factors associated with youth smoking"? The authors need to focus on one topic. If the topic is on the factor associated with youth smoking, it seems to lack of originality. If it is on trend of youth smoking between 1995 and 2015 in Ireland, the authors should analyze and explain the causes of the decline smoking rate to find the originality. 2. If the main results of this paper are to show a decreasing trend in youth smoking rate in Ireland, especially among girls, the authors should explain the factors that contributed to reducing the prevalence rate of youth smoking and explain why the decreasing trend was shown in girls. Delete the analysis of the risk factors for smoking by pooling the data for three years. Instead, analyze the 2007, 2011, and 2015 data separately and analyze the changes in risk factors for smoking. 3. The authors explained the reason why adolescents' smoking rate have declined in discussion section. However, there is no mention why the girls' smoking rate was decreased faster than boys. It must be interpreted in discussion section. 4. If prevalence rate is decreased, the numbers of smoking friend is decreased naturally. Therefore, It is not appropriate to explain the reason why the smoking prevalence was decreased due to the fact that smoking friend was decreased. Therefore correct it appropriately. 5. Family structure and truancy were also changed in Ireland during this period. These changes can be explained as important variables if the changes were affected in reduction of smoking rate. But If not, it will be deleted.
--

VERSION 2 – AUTHOR RESPONSE

Responses to Reviewers' Comments:

Reviewer: 1

Reviewer Name: Kristiina Patja

Institution and Country: Pro Medico, Association for continuous professional development in Finland

Competing Interests: None declared

Thank you very much for your constructive and encouraging comments.

We have made a reference to our previous electronic cigarettes publications in the introduction as suggested.

Sorry we have tried to improve the titles by revising all 4 legends:

Table 1 Sample sizes, gender and response rates of the ESPAD Ireland surveys (1995 -2015)

Table 2 Summary of the results and changes in key measures associated with smoking in Irish ESPAD surveys 2007-2015.

Table 3 30-day smoking prevalence (%) in Irish ESPAD surveys from 1995-2015

Table 4 Multivariate logistic regression results of various factors potentially associated with smoking from Irish ESPAD surveys 2007-2015

The effects of the severe recession seen in Ireland on smoking in adolescences is unclear but the decline continued and does not seem to have accelerated or slowed. The perception of relative 'well offness' decreased in 2011 when the recession was at its worst but increased in 2015 when things had begun to improve suggesting that the perception of the relative decline and recovery was uneven.

We have revised the conclusions as also suggested by another reviewer but did not think we should repeat too much of the discussion. We are hoping the results speak for themselves but wanted to offer some advice in the recommendations.

Reviewer: 2

Reviewer Name: Sofia Ravara

Institution and Country: Faculty of Health Sciences, University of Beira Interior and CHCB University Hospital, Covilhã, Portugal.

Competing Interests: None declared.

Thank you very much.

Reviewer: 3

Reviewer Name: Arja Rimpelä

Institution and Country: University of Tampere, Finland Competing Interests: None declared

Thank you, point taken re dates, dates now included on page 6 row 27 as suggested.

The word 'increased' has been removed from Discussion page 14 row 25 as suggested.

Reviewer: 4

Reviewer Name: Hyeon Woo Yim

Institution and Country: Department of Preventive Medicine, College of Medicine, Catholic University of Korea, South Korea Competing Interests: None declared

1. We are documenting the very rapid decline in smoking in adolescences in Ireland, which has very strong Tobacco Control interventions, and the unusual gender aspects of our adolescent smoking situation. We are limited in what we can do by the data available which allows us to look at some of the personal and family factors associated with smoking in a regulated Tobacco Control scenario.
2. We assume from the literature and established research and our own previous research (www.tri.ie) that policies included in the WHO FCTC, which have been applied in Ireland, more than in most other countries, result in reductions of smoking in adults and adolescences. The reductions in prevalence however, with some exceptions e.g. price elasticity of demand, are not easily predicted and vary to some extent in different countries. It was for this reason that the changes in Ireland, which are very marked, are reported. We are trying to add to the understanding of the known legislative factors at play in Ireland and also the family and personal factors that may be of influence in the past and future decline in smoking in adolescents in Ireland and in general.
3. We are struggling to understand this fully and are actually working on it at present in another project but are reluctant to speculate without adding data. It seems to us that some of the legislative interventions e.g. Smokefree Legislation and Point of Sale bans made have been more effective in girls than boys but would prefer not to add this to the discussion at present, if possible, as it is not reasonable to make this assertion from these ESAPD data alone but we have mentioned our problem now in the discussion.
4. We are not too sure about this because it is not always clear whether adolescences only smoke because their friends smoke or become friends with other people who smoke only because they smoke or cease to be friends with those established friends who give up smoking is not always clear. It should be possible for smokers to continue to have the same number of friends who smoke if they wished unless there were no other smokers in their population. We will however make every effort not to mislead and hope that you can accept that our changes help in that regard and we have added a reference to help clarify this point. (Stewart-Knox BJ, Sittlington J, Rugkåsa J, et al. Smoking and peer groups: results from a longitudinal qualitative study of young people in Northern Ireland. *Br J Soc Psychol* 2005;44(3):397-414.)
5. Yes, there was significant change in truancy, but insignificant change in family structure (i.e. one-parent family). Logistic regression shows significant smoking relationships with truancy. As to family structure, it shows significant smoking relationship to female students.

We have clarified the changes in these variables in Table 2 by adding an asterisk to the significant changes. Most variables have changed across the survey waves. Only two variables did not change significantly, which are “risk of smoking cigarettes occasionally” and “One-parent family”.

“Risk of smoking cigarettes occasionally” is shown to be strongly associated with smoking. Therefore, we would still have thought it valid to include them as they are personal factors which could be changed and help reduce smoking further. Obviously if they are not significantly different they would not be considered to have contributed to the decline in smoking and would not then have helped explain the observed reductions and we have clarified this in the text. They could still be important interventions in trying to achieve a ‘tobacco free society’ and were reported for that reason.

In the regression, all variables listed in Table 2 are significantly related to smoking, either for both genders or for one gender only. Therefore, we feel all variables in Table 2 should still be included.

6. We have revised the conclusions to explain the links to the data more clearly as suggested.

VERSION 3 – REVIEW

REVIEWER	Gyeon Woo Yim Department of Preventive Medicine, College of Medicine, Catholic University of Korea
REVIEW RETURNED	20-Feb-2018
GENERAL COMMENTS	I have no further comments to add.

VERSION 2 – AUTHOR RESPONSE

VERSION 3 – REVIEW

REVIEWER	
REVIEW RETURNED	

GENERAL COMMENTS	
--

REVIEWER	
REVIEW RETURNED	

GENERAL COMMENTS	
--

REVIEWER	
REVIEW RETURNED	

GENERAL COMMENTS	
--

REVIEWER	
REVIEW RETURNED	

GENERAL COMMENTS	
------------------	--

VERSION 3 – AUTHOR RESPONSE

VERSION 4 – REVIEW

REVIEWER	
REVIEW RETURNED	

GENERAL COMMENTS	
------------------	--

REVIEWER	
REVIEW RETURNED	

GENERAL COMMENTS	
------------------	--

REVIEWER	
REVIEW RETURNED	

GENERAL COMMENTS	
------------------	--

REVIEWER	
REVIEW RETURNED	

GENERAL COMMENTS	
------------------	--

VERSION 4 – AUTHOR RESPONSE

VERSION 5 – REVIEW

REVIEWER	
REVIEW RETURNED	

GENERAL COMMENTS	
------------------	--

REVIEWER	
REVIEW RETURNED	

GENERAL COMMENTS	
--

REVIEWER	
REVIEW RETURNED	

GENERAL COMMENTS	
--

REVIEWER	
REVIEW RETURNED	

GENERAL COMMENTS	
--

VERSION 5 – AUTHOR RESPONSE